# Driving Chromatin Organisation through N6-methyladenosine Modification of RNA: What Do We Know and What Lies Ahead?

**DOI:** 10.3390/genes13020340

**Published:** 2022-02-12

**Authors:** Tommaso Selmi, Chiara Lanzuolo

**Affiliations:** 1Consiglio Nazionale delle Ricerche, Istituto di Tecnologie Biomediche, Via Fratelli Cervi 93, 20054 Milano, Italy; chiara.lanzuolo@cnr.it; 2Istituto Nazionale di Genetica Molecolare, Via Francesco Sforza 35, 20122 Milano, Italy

**Keywords:** N6-methyladenosine, chromatin, histone modifications, transposable elements, chromatin-associated RNAs, transcription, LLPS (Liquid–Liquid Phase Separation)

## Abstract

In recent years, there has been an increase in research efforts surrounding RNA modification thanks to key breakthroughs in NGS-based whole transcriptome mapping methods. More than 100 modifications have been reported in RNAs, and some have been mapped at single-nucleotide resolution in the mammalian transcriptome. This has opened new research avenues in fields such as neurobiology, developmental biology, and oncology, among others. To date, we know that the RNA modification machinery finely tunes many diverse mechanisms involved in RNA processing and translation to regulate gene expression. However, it appears obvious to the research community that we have only just begun the process of understanding the several functions of the dynamic web of RNA modification, or the “epitranscriptome”. To expand the data generated so far, recently published studies revealed a dual role for N6-methyladenosine (m6A), the most abundant mRNA modification, in driving both chromatin dynamics and transcriptional output. These studies showed that the m6A-modified, chromatin-associated RNAs could act as molecular docks, recruiting histone modification proteins and thus contributing to the regulation of local chromatin structure. Here, we review these latest exciting findings and outline outstanding research questions whose answers will help to elucidate the biological relevance of the m6A modification of chromatin-associated RNAs in mammalian cells.

## 1. RNA Modifications 

More than 100 RNA modifications have been reported in the transcriptome of organisms spanning from archaea to eukaryotes [1]. These chemical alterations of RNA nucleotides expand the properties of a given RNA sequence, thus affecting its function [2]; however, the precise effect of each modification on distinct RNAs and the consequences on genome function are still the subjects of intensive research. In experimental models, RNA modifications appear dynamically regulated, primarily as a form of adaptation to stress. Modifications are present at high frequency in abundant and stable RNA molecules, such rRNAs and tRNAs, and to a lower extent in other RNAs, such as mRNAs, and other non-coding RNAs such as long non-coding RNAs (lncRNAs) and small nuclear RNAs (snRNAs) [3,4]. The depletion of the enzymes belonging to the RNA modification machinery shed partial light on the biological function of a few of these modifications, and genetic studies uncovered a robust association between mutations of RNA modifying enzymes, developmental defects [5], and cancer [6].

### 1.1. N6-methyladenosine in Mammals

N6-methyladenosine (m6A) is the most abundant internal mRNA modification, occurring at a frequency of 1–3 nt/1000 nt, [7,8], mostly in the coding sequence and in the 3’ untranslated region (3′UTR), and is significantly enriched around stop codons at the consensus motif RRACH (R = A or G, H = A, C, U) [9,10,11]. Data suggest that m6A deposition happens co-transcriptionally in the nucleus [12,13] and is catalysed by the methyltransferase complex (MTC) [14,15], which is composed by a core catalytic heterodimer including METTL3 (methyltransferase-like 3) and METTL14 (methyltransferase-like 14) [14,16], and additional factors (WTAP, ZC3H13, RBM15/RBM15B, VIRMA) with regulatory functions [14,17,18,19,20]. METTL3 knockout mice display embryonic lethality, as the METTL3 protein regulates naïve pluripotency genes, and its depletion leads to imperfect termination of the naïve state [21].

In addition to mRNAs, m6A also occurs in non-coding RNAs, such as a group of primary microRNAs (pri-miRNAs), where it has been shown to regulate their processing [22]; in lncRNAs, where it has been shown to regulate functional activities [23,24]; and in the non-coding small nuclear RNA (snRNP) U6 [25,26], potentially affecting pre-mRNA splicing. Data suggest that METTL3 and METTL14 are responsible for most of the poly(A) mRNA m6A deposition, while METTL16 (methyltransferase-like 16) [27] appears to mostly target structured RNAs contained in intronic sequences and non-coding RNAs, [25], although not exclusively [26,28]. Similar to what has been observed with METTL3, knockout attempts of METTL6 in mice resulted in embryonic lethality [29].

A group of five m6A reader proteins named the YT521-B homology (YTH) domain-containing proteins (YTHDC1, YTHDC2, YTHDF1-3) directly bind to m6A methylated RNAs through the highly conserved YTH domain [30,31]. Of the 5 members of this family, YTHDC1 is involved in mediating splicing regulation and trafficking in the nucleus [32,33], while YTHDC2 binds to m6A residues in mRNAs of germ cells, affecting their maturation [34]. Knockout of YTHDC1 is embryonically lethal, and its depletion from mouse male/female germ cells impairs their maturation [35]. On the other hand, YTHDC2 knockout mice are viable but infertile [36]. With regard to YTHDF1-3, these were originally identified with diverse functions affecting mRNA translation or turnover [37,38]; however, recent work suggests that YTHDF1-3 might preferentially regulate mRNA turnover, with some degree of functional redundancy [39,40].

Additional readers, hnRNPG [41], hnRNPC [2], hnRNPA2/B1 [22] (heterogeneous nuclear ribonucleoproteins), and IGF2BP (insulin-like growth factor binding protein) [42], all lacking the m6A-binding YTH domain, appear to mediate m6A-dependent functions, such as transcripts’ splicing and turnover. Given the ability of m6A to destabilise RNA hairpins [43,44] to expose RNA binding motifs, these readers might be recruited on m6A-RNAs by changes in the methylated RNA’s secondary structure [2,43,44,45].

The two only known m6A demethylases belong to the Fe(II)/α-ketoglutarate-dependent dioxygenase family (ALKB) [46]. The discovery of such specialised m6A “erasers” has been fundamental to the definition of the dynamic nature of the m6A “epitranscriptome”. The “eraser” enzyme ALKBH5 (ALKB homolog 5) is specifically active on the m6A modification [47] and is up-regulated in some cancers, where it promotes the stabilisation of oncogenes during hypoxia [48]. The other known m6A demethylase is FTO (fat mass and obesity associated gene). Despite having been originally characterised as the main m6A demethylase [49], FTO shows stronger enzymatic activity towards N^6^,2′-O-dimethyladenosine (m^6^Am), especially in snRNAs [50]. FTO knockout causes splicing defects [51] and acts as an oncogene in acute myeloid leukaemia [52].

Overall, m6A mainly regulates gene expression by affecting mRNA stability and promoting its turnover [37,53,54,55], however, published literature also suggest that m6A-carrying mRNAs undergo increased cap-dependent translation [56], and data obtained under stress conditions support a model where the m6A modification of 5’UTR (5’ untranslated region) mediates cap-independent translation [57,58]. Finally, m6A could regulate splicing of a group of mRNAs by the interaction of the nuclear m6A reader YTHDC1 with members of the SRF (serine/arginine (SR)-rich proteins) protein family in nuclear speckles [32]. 

The genetic perturbation of the m6A machinery (writer, reader, and eraser genes) in different experimental models affects a striking variety of molecular processes including, but not restricted to: adipogenesis [59], spermatogenesis [15,34], oocyte maturation [35,60], neurodevelopment [61,62,63], pluripotency [21,54,55], circadian rhythm regulation [64], senescence [65], multiple cancerous traits [52,66,67,68,69,70,71,72], and antitumor immunity [73,74].

### 1.2. m6A and Epigenomic Regulation

The phenotypic pleiotropy [75] that derives from the experimental perturbation of the m6A machinery has been ascribed to the central role that this modification plays in the cell. This could potentially point to the limitations of classic genetic experimental approaches for the study of cellular modification shared by so many targets [75]. In light of the recent reports reviewed here, the variety of phenotypes associated with m6A-preturbations could, at least partly, depend on the direct action of m6A on the epigenome. In support of an epigenetic role of m6A, previous literature shows that METTL3-dependent m6A regulates X-chromosome inactivation via a group of methylated adenosine residues on theXIST (X-inactive specific transcript) lncRNA that are bound by the nuclear m6A reader YTHDC1 [23]. Furthermore, in multiple human breast cancer cell lines, m6A modification of nucleotide A783 of the developmentally regulated HOTAIR lncRNA (HOX antisense intergenic RNA) determines the loss of the lncRNA’s repressive function, promoting proliferation and invasiveness [76]. Lastly, in mouse neural stem cells, the genetic depletion of m6A writers has been found to enhance the degradation of mRNAs encoding for acetyltransferase EP300 (E1A-associated protein p300) and CBP (CREB-binding protein), hence regulating the abundance of activating histone modifications [62]. For the keen reader, the epigenetic effects of m6A have also been summarised recently by Kan and colleagues [77] and Zhao and colleagues [78]. Table 1 outlines examples of the interplay between m6A and chromatin accessibility.

## 2. Transposable Elements, RNA Modifications, and Chromatin Organisation

The human and murine genomes are characterised by the presence of highly repetitive elements (RE), contributing to an estimated 50% and 70% of their genetic material, respectively [88]. Transposable elements (TE) are a class of RE that integrate in the host genome through either RNA or DNA intermediates. Among the RNA retrotransposons, LTR derive from ancient retroviruses, thus they encode for viral ORFs and carry 5’ and 3’ long terminal repeats (LTR). Even though they are mostly inactive, the youngest evolutionary LTR families, such as ERV-I, ERV-II, and ERV-K (endogenous retroviruses) in humans, and intracisternal A particle (IAP) elements in mice, are capable of retrotransposition [88]. Moreover, non-LTR retrotransposons are also capable of autonomous integration and are referred to as long interspersed nuclear elements (LINEs) [89]. To prevent random integration and interference with gene transcription networks, TEs are subject to constant silencing, achieved by epigenetic mechanisms such as histone H3K9me3 and DNA methylation, and RNA interference by small RNAs, such as piRNAs (Piwi interacting RNAs). TEs are thought to play a key role in the evolution of the genome, and TE distribution into the genome is not random; in fact, they are arranged in domains of heterochromatin [90,91]. Interestingly, similar to what happens to RNA modifications, the activation of TEs in the eukaryotic genome can be triggered by stress conditions [92]. 

Evidence that RNA modifications can control TE expression comes from earlier studies in the Drosophila model, where DNMT2, a tRNA 5-cytosine methyltransferase (m5C), appears to contribute to the inhibition of TE re-activation [93]. Such findings have recently been confirmed and more thoroughly investigated by Genenncher and colleagues, who demonstrated that KO of both the tRNA m5C transferases NSUN2 and DNMT2 promotes TE re-activation and genomic instability [94]. The authors hypothesise that NSUN2 and DNMT2 achieve TE silencing by promoting the translation of chromatin remodelling complexes, rather than by direct modification of the TE. Even though more work is needed to draw robust conclusions on the molecular mechanisms involved, these studies point toward an active interplay between the RNA modification machinery and the maintenance of genomic stability via the regulation of TE.

### 2.1. IAP and LINE Regulation by m6A and Histone Modifications in Mouse Embryonic Stem Cells

Transcriptional re-activation of certain families of TEs takes place during the epigenetic reprogramming of mESc [95,96]. Although originally thought to be a consequence of increased chromatin accessibility during embryonic reprogramming, the re-activation of LINE-1 elements is now considered as an integral part of this process [96]. In fact, timely LINE1 activation ensures the correct resolution of early transcriptional programs. Moreover, different studies have shown that TEs, including IAPs, are kept inactive by mechanisms that are alternative to DNA methylation [97,98,99].

Three studies have recently reported the regulatory activity of m6A over LINE1 in mouse ES cells [79,80,81]. The first study [79], which looked at genome-wide regulation of chromatin accessibility by m6A (Figure 1A), found the robust enrichment of m6A at chromatin-associated repetitive RNAs (carRNAs) of the mESc genome. The LINE1 family of carRNAs showed the most significant reduction in m6A after METTL3 KO, coupled with increased stability. This correlated with genome-wide enhanced chromatin accessibility and the deposition of active histone marks such as H3K4me3 and H3K27ac, possibly mediated by the action of EP300 (histone acetyltransferase p300) and YY1 (Yin Yang 1 transcription factor). Biologically, the modulation of the methylation levels of LINE1 in METTL3 KO mESc affected self-renewal and differentiation potentials [79].

In a complementary study [80], Liu and colleagues focused on the role of the nuclear m6A reader YTHDC1 in silencing TEs (LINE1, ERVK, IAPs) (Figure 1B). The authors found that YTHDC1 acted by reducing the levels of m6A-marked TE RNAs through direct binding. The RNA-bound YTHDC1 recruited the histone methyltransferase SETDB1 (SET domain bifurcated histone lysine methyltransferase 1) to promote the deposition of H3K9me3 at the corresponding loci, in order to repress chromatin accessibility. Biologically, the m6A binding activity of YTHDC1 was key in preventing the activation of a stem cell-specific 2C-like transcriptional state [80]. In the absence of YTHDC1, m6A-marked LINE1 was recruited on the Dux1 locus where it mediated chromatin repression [80]. These findings were confirmed in a follow up study by Chen et al., who found that m6A-marked LINE1 can complex with Nucleolin (NCL) to inhibit the re-expression of TE, controlling the execution of the 2C-like state program [81] (Figure 1C).

The m6A machinery appears to be necessary to maintain heterochromatin marks at the intracisternal A particle (IAP)-type family too, as reported by two studies [82,83]. In their work, Xu and colleagues [82] observed the METTL3-dependent deposition of repressive H3K9me3 and H4K20me3 at IAP elements, and m6A-dependent IAP RNA de-stabilisation. According to their model, METTL3 mediated the methylation of IAP RNAs and recruited YTHDC1. Importantly, for their repressive function, m6A-marked IAP RNAs were retained at their chromosomal loci. The METTL3–YTHDC1 subsequently recruited SETBD1 and TRIM28 (tripartite motif containing 28) to repress chromatin accessibility via histone methylation (Figure 1B). Concomitantly, Chelmicki and colleagues [83] published very similar findings while performing a whole-genome CRISPR screen for the identification of novel negative regulators of ERV activation. Using KO cells and auxin-inducible degron constructs (AID, which allow for an acute and reversible depletion of the METTL3 protein), the authors confirmed the presence of m6A in IAP RNAs; moreover, they found that the reader YTHDF2 binds to methylated IAPs to promote their turnover (Figure 1B). Accordingly, the authors found increased levels of IAPs in m6A-depleted cells, however, they noted that other TEs like LINEs, which also carry METTL3-/METTL14-dependent m6A, appeared to be stabilised by m6A, and might therefore be regulated by other m6A readers. Importantly, the authors focused on the acute effects of m6A depletion and showed that IAPs were quickly degraded in response to the deposition of m6A, although not concomitantly with the formation of heterochromatin at the corresponding genomic loci. This study employed AID, an important experimental alternative to genetic KO, whose results suggest that histone modifications observed in genetic KO models could arise from the process of cellular adaptation to the long-term depletion of m6A, but might not be readily installed upon loss of the modification to the IAP RNAs [83].

### 2.2. MSR Repeats Turnover and RNA:DNA Hybrid Formation by m6A

Major satellite repeats (MSR) are repeated DNA elements of constitutive heterochromatin spanning the peri-centromeric areas of the chromosome [100]. A portion of these sequences are transcriptionally active, and when transcribed they form RNA:DNA hybrids (R loops) that recruit HP1 (heterochromatin protein 1) and the SUV39 (suppressor of variegation 3–9) histone methyltransferase to maintain the local H3K9me3 and compact chromatin structure, ultimately ensuring the maintenance of genome integrity [101,102]. R loops are RNA:DNA hybrids that form between the nascent mRNA and the DNA, mainly during transcription and in response to DNA damage. The role of R loops in controlling gene expression and genomic integrity has been demonstrated in a number of mammalian models [103], and in some instances, m6A of RNAs in R loops exerts control over their formation and resolution. For example, in cancer cell lines, METTL3-dependent m6A promotes the formation and stability of R loops at genomic sites of double-strand breaks through the binding of YTHDC1 [104]. On the other hand, in pluripotent stem cells, the binding of YTHDF2 to R loops that are m6A-marked triggers their degradation [105]. So far, data support the existence of a connection between m6A and R loops, however, the details of the regulatory effects of m6A over these structures (and vice versa) are not clear, as these might depend on the cell context and/or on the specific R loop trigger event.

In their study, Duda and colleagues investigated the potential involvement of m6A in the process of MSR silencing [84]. They showed that MSR were targeted by METTL3/METTL14 in vitro, and that m6A modification of MSR decreased in METTL3 and METTL14 KO mES cells. M6A of MSR RNAs promoted their association to chromatin and the generation of RNA:DNA hybrids. The study also provided a quantification of the methylated fraction of MSR repeats and LINE1 (20% and 70% of transcripts, respectively), and reported the stabilising effect of m6A on these two families. However, the methylation of MSR and LINE1 did not entirely depend on METTL3/METTL14, as KO cells showed residual methylation of these RNAs. The authors did not investigate the mechanism of MSR and LINE1 stabilisation in their mouse ES cells models in detail, although they suggest that neither YTHDC1 nor YTHDC2 showed increased in vitro affinity for m6A-MSR, hinting to the presence of other unidentified factors that can bind these two classes of m6A modified RNAs [84].

Another interesting observation from this study is the quantification of both m5C and m6A on the bulk of chromatin-associated RNAs by two methods (LC-Mass spectrometry and m5C/m6A meRIP). This showed that m5C is only present at low intensities in this fraction, as opposed to significantly higher m6A levels [84]. Due to its nearly background level, the authors have excluded a regulatory role for m5C. This is an interesting observation, as earlier studies reported that m5C might regulate local chromatin organisation by the methylation of nascent mRNAs [106] and enhancer RNAs (eRNAs) [107]. It is well described that epigenetic mechanisms are genomic-site-dependent; therefore, based on this data, it cannot be excluded that m5C or other modifications might have regulatory roles on subsets of specific RNAs at different chromatin loci, or perhaps even cooperate with m6A. With regards to this possibility, the interplay between m6A and m5C has been reported to enhance the translation of the mRNA of p21 in the cytoplasm [108].

Lastly, the authors observed no significant alteration to H3K9me3 levels or HP1 localisation in METTL3 and METTL14 KO cells [84], as opposed to what has been reported by the other studies that focused specifically on LINE and IAP elements after the genetic knockout of MTC components or of the reader YTHDC1 [79,80,81,82].

### 2.3. Summary of: Transposable Elements, RNA Modifications, and Chromatin Organisation

The reviewed data provides robust evidence that m6A ensures the correct execution of the differentiation programme of mESc by preventing the uncontrolled re-activation of transposable elements (IAPs, LINE1). This relies on the nuclear crosstalk between m6A of TE-RNAs and repressive epigenetic factors. Within this circuit, two levels of TE-RNA repression could co-exist: (1) the m6A-dependent recruitment of TRIM28 and SETDB1, exerted by METTL3 and YTHDC1, which is critical to the establishment of a H3K9me3 repressive environment [80,81,82], and (2) the degradation of m6A-marked TE transcripts involving the readers YTHDC1 and YTHDF2 [79,83]. In experimental settings however, some classes of TEs bound by YTHDC1 were destabilised by m6A modification (IAPs) while others were stabilised or showed opposite trends in similar studies, suggesting that an additional m6-dependent regulation of TEs takes place. This could be dependent on the activity of additional m6A methyltransferases (METTL5, METTL16, and ZCCHC4) [26,109,110] as suggested by Duda and colleagues [84], or depend on the activity of reader proteins, which could be tuned by post-translational modifications or other molecular interactions. 

## 3. Chromatin Accessibility, Transcriptional Regulation, and m6A 

The link between m6A and accessible, transcribed chromatin has been established by different studies. In particular, according to Slobodin and colleagues, the deposition of m6A happens co-transcriptionally and the processivity rate of RNA polymerase II affects the overall level of transcript m6A methylation and its translation [12]. The METTL3 association with chromatin is a process that is also regulated by heat shock, leading to the methylation of specific transcripts during hypothermia [13]. In human leukaemia cells, the CEBPz (CAATT-enhancer binding protein z) recruits METTL3 onto the promoter of oncogenic transcription factors to increase the translational efficiency of the methylated transcripts [67]. In human pluripotent stem cells, SMAD2 and SMAD3 promote the loading of MTC on the pluripotency factor gene *NANOG*, effectively enabling the degradation of the *NANOG* transcript and a timely exit from pluripotency [111].

The recruitment of MTC to chromatin is a key regulatory step in the m6A functional cascade, and a group of studies provided new insights on this while reporting on the epigenomic–epitranscriptomic crosstalk. These studies focused on the activity of m6A in gene bodies [85,86,87], and intergenic loci [79] in mESc and human immune cells.

The study by Liu and colleagues, which unveiled the regulatory role of m6A over LINE1 [79], sought to identify the role of m6A modification of carRNAs (promoter-associated RNAs, enhancer RNAs and repetitive RNAs) in shaping genome-wide chromatin accessibility. The authors found that chromatin accessibility was increased in METTL3 KO (genome-wide), together with the acquisition of the accessible histone marks H3K4me3 and H3K27ac (Figure 1A). METTL3 KO triggered transcription, especially from the genomic loci downstream from intense m6A peaks, and increased the turnover rate of a fraction of the methylated carRNAs, and was mediated by the reader YTHDC1, in conjunction with the nuclear exosome complex (NEXT). Site-directed de-methylation of specific carRNAs by dCas13-FTO in WT mESc recapitulated the effects of METTL3 depletion on histone marks and transcription, thus supporting a role for m6A of carRNAs in repressing chromatin accessibility and transcriptional output [79].

Li and colleagues subsequently showed that m6A deposition in nascent mRNAs promotes genome-wide de-methylation of the repressive H3K9me2, and feedbacks positively on gene expression [85]. The authors collected preliminary evidence of the direct regulation of m6A over H3K9me2 using a reporter model in HEK293, and later confirmed their findings in mESc. Mechanistically, they found that the nuclear reader YTHDC1 recognised and interacted with m6A-methylated nascent transcripts to recruit lysine demethylase KDM3B (Figure 2A), which in turn demethylated H3K9me2. The findings were supported by targeting dCas13-YTHDC1 on specific genomic loci, where the authors observed an enhanced recruitment of KDM3B and concurrent demethylation of H3K9me2. These data show that m6A of nascent mRNAs can directly promote the de-methylation of histone tails, in order to favour chromatin accessibility and increase the transcriptional output of mRNAs [85]. As observed in the previous study from Liu and colleagues, the reader YTHDC1 appeared to mediate the action of m6A on chromatin accessibility.

While the previous two studies highlighted an active role for m6A and its reader YTHDC1 in guiding histone modifications and chromatin accessibility, albeit with opposite functions, Huang and colleagues described a mechanism where a specific histone modification promotes m6A deposition on nascent RNAs [86]. The authors uncovered the key interaction between the m6A writer METTL14 and H3K36me3, which marks actively transcribed regions. This is key in assuring a functional and possibly physical interaction between MTC and RNA polymerase II. The depletion of H3K36me3 (either by KO of histone lysine methyltransferase SETD2, or overexpression of histone lysine demethylase KDM4A) triggered a reduction in m6A levels. This was a unidirectional effect, as the depletion of m6A through METTL14 KD did not alter H3K36me3 levels. The authors confirmed their findings on selected transcripts, such as MYC and ACTB, where guiding of dCas9-KDM4A (MYC) or dCas9-SETDB2 (ACTB) reduced and increased m6A levels, respectively. Biologically, the decrease of m6A, dependent on H3K36me3 depletion, interfered with mESc differentiation, which is consistent with the known role of m6A in promoting pluripotency genes’ turnover [71] (Figure 2B). 

Finally, Wu and colleagues reported in human immune cells that histone lysine demethylase KDM6B [87] recruited MTC on specific gene subsets. In a model of infection, the repressive mark H3K27me3 acted as a barrier to m6A modification, and KDM6B acted by demethylating this repressive mark to support the m6A modification of nascent transcripts by recruiting MTC. YTHDF2 negatively regulated the stability of the m6A-marked KDM6B transcript. In YTHDF2 KO cells, decreased levels of H3K27me3 led to the up-regulation of cytokine mRNA transcription (Il-6), and also allowed the deposition of m6A on mRNAs coding for histone modifiers, and both actions were mediated by KDM6B. Interestingly, the authors reported that in murine cells, YTHDF2 did not bind to KDM6B mRNA, suggesting that mouse and human RNA targets of this reader might not fully overlap [87].

### Summary of: Chromatin Accessibility, Transcriptional Regulation, and m6A 

While it cannot be excluded that inherent limitations of the experimental approaches (Box 1) could be biasing data interpretation, these works [79,85,86,87], together with early reports [23,62], build a robust case for considering m6A an integral player in the epigenetic circuit. The ability shown by both reader and writer proteins to interact with epigenetic players, either at the mRNA or protein level, might be the obvious justification for the variety of effects that have been reported. In loci where active transcription takes place, m6A could function by promoting chromatin openness and transcriptional output [85], while in some cases, specific accessible histone marks (H3K36me3) would recruit m6A to provide some form of post-transcriptional control over the transcribed genes, with no direct impact over the local chromatin state [86]. In intergenic regions, however, m6A decorates other classes of RNAs, such as carRNAs, and it appears to repress accessibility and transcription [79]. These interesting observations would suggest that the m6A machinery functionally integrates within the local chromatin context. How this is regulated needs to be clarified, but surely, given the vast amount of coding and non-coding RNAs targeted by m6A, the active role of m6A within both activator and repressive chromatin environments is not surprising. On a higher level of regulation, how the m6A-dependent methylation of nascent transcripts and the m6A-dependent epigenetic regulation of the chromatin state are functionally integrated is still underexplored.

Box 1Technical approaches to the study of m6A epigenomic crosstalk.On a technological standpoint, the study of the crosstalk between m6A and the histone signature relies on achieving high resolution mapping of known modifications, both in the genome and in the transcriptome. That is why approaches such as CHIP-Seq (Chromatin Immunoprecipitation-Sequencing) to map histone tail modifications and meRIP-Seq (m6A-specific methylated RNA immunoprecipitation-Sequencing) to map m6A on the RNA fraction have been employed by most of the reviewed studies [62,79,80,82,83,85,86,87]. Fractionation approaches aimed at separating the cytoplasm from the nucleus aided in retrieving RNAs specifically associated to the chromatin portion and such RNAs were later analysed either by HPLC-MS (liquid chromatography coupled with mass spectrometry), or by meRIP-seq [79,84,85,86]. One study [82] attempted the validation of selected methylated adenosines on a target RNA at single nucleotide resolution, by employing SELECT (single-base elongation and single-base ligation qPCR amplification method) [112]: this provides a method for validating methylated residues in support of mechanistic insights. The reviewed studies perturbed the m6A machinery by employing approaches such as genetic knock out, as well as acute depletion via the AID (Auxin Inducible Degron) system of reader and writer proteins. As regards to this, data show that acute depletion methods might aid scientists who wish to focus on a time restricted, dynamic aspect of the epigenomics-epitranscriptomics crosstalk [79,83]. In order to gain mechanistic insight, some of the studies investigated the interaction of modified RNAs with the chromatin and their relative genomic location by employing methods such as CHIRP-Seq (Chromatin Isolation by RNA Immunoprecipitation-Sequencing) [80,82], which in one case was validated using a published GRID-Seq (Global RNA Interactions with DNA sequencing) dataset [113]. Finally, the impact of m6A over the activity/fate of selected repetitive elements was validated by targeted RNA de-methylation, using an ALKBH5/FTO enzyme tethered to a dCas13, a very precise method for achieving precise local disruption of the m6A network, without the need for the genetic manipulation of the target transcripts [79,85]. A similar method, based on dCas9 has also been used to guide histone modification proteins or m6A readers to the chromatin region overlapping with m6A peaks, to investigate the effect of histone modulation over the local abundance of m6A and vice versa [82].

## 4. m6A and Liquid–Liquid Phase Separation

With regards to the execution of context-specific nuclear functions, an emergent field of investigation is the assembly of membraneless biomolecular condensates via the process of liquid–liquid phase separation (LLPS). Biomolecular condensates are made of a liquid and a dense phase, where the dense phase acts by concentrating and organising functionally related molecules to support the execution of biological processes [114]. The maintenance of the genomic structure and its function in the nucleus depends on the structural organisation provided by LLPS [115]; in fact, the occurrence of LLPS is key in physiological [116] and diseased states [117]. RNAs are key triggers of LLPS, and features such as their nucleotide composition, length, structure, and expression levels all affect physical properties of the membraneless condensates [118].

RNA modifications can affect LLPS in the cytoplasm, as m6A-marked mRNAs are sorted to stress granules, p-bodies, and neural mRNA granules through the binding of cytoplasmic YTHDF readers [119]. An important feature of mRNAs that favours LLPS is the occurrence of clusters of methylated nucleotides in a single mRNA molecule, which increases the likelihood of its separation in dense droplets through the recruitment of multiple YTHDF readers [119,120,121]. In addition to YTHDFs, the nuclear reader YTHDC1 has also recently been observed to undergo LLPS in human myeloid leukaemia cells [122]. According to this study, m6A and YTHDC1 appear to support the cancerous phenotype of AML cells by forming nuclear condensates (nYACs) that protect the mRNAs’ coding for oncogenes from degradation by the PAXT (polyA tail exosome) complex. Interestingly, nYACs seem to preferentially occur in AML cells as opposed to non-leukemic hematopoietic cells, and their formation is dependent on the presence of an N-terminal IDR (intrinsically disordered region) on the YTHDC1 protein [122].

Overall, there is growing evidence that RNA m6A is a positive regulator of LLPS, and that it can promote the assembly of YTHDC1 biomolecular condensates in the nucleus. Based on the data reviewed here, it is tempting to speculate that m6A of chromatin-associated RNAs could be driving LLPS of epigenetic regulators, both in euchromatin and heterochromatin domains.

## 5. Conclusions

These exciting novel data unveil a previously unappreciated crosstalk between epitranscriptomics and epigenomics and pose the foundation for its deeper investigation, especially in human cellular models. The future studies of such crosstalk will improve our understanding of how the m6A modification of RNA integrates in the nuclear regulatory mechanism, and will hopefully provide new insights on how to modulate gene expression, either experimentally or therapeutically.

## Figures and Tables

**Figure 1 genes-13-00340-f001:**
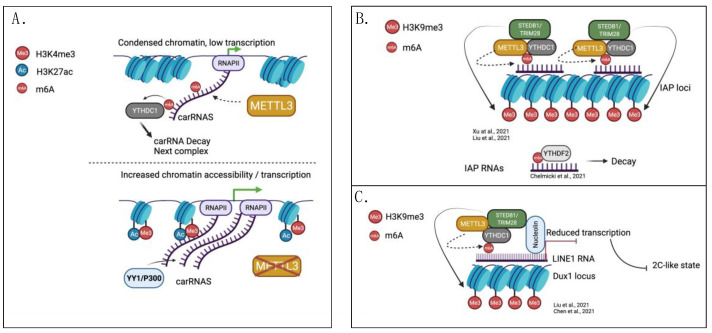
The regulation of transposable elements and chromatin accessibility by m6A. (**A**) METTL3-dependent methylation of carRNAs maintains condensed chromatin at intergenic regions and promotes carRNAs degradation by the NEXT (nuclear exosome targeting) complex. Loss of carRNA methylation leads to increased chromatin accessibility and enriched transcription, associated with increased histone H3-lysine4 trimethylation (H3K4me3) and histone H3-lysine27 acetylation (H3K27ac). carRNAs can now recruit epigenetic factors such as YY1 and EP300 to maintain an open chromatin conformation and downstream transcription. (**B**) METTL3 deposits m6A on intracisternal A particle (IAP) RNAs. YTHDC1 recognises and binds to methylated IAPs, and in conjunction with METTL3 recruits the histone methyltrasferase SETDB1 and its co-factor TRIM28. This complex establishes histone H3-lysine9 trimethylation and maintains a closed chromatin conformation at IAP loci. This leads to an overall reduction in the transcription of IAP RNAs. m6A-marked IAP RNAs are degraded by YTHDF2. (**C**) m6A-marked LINE1 silences the Dux1 locus in mouse embryonic stem cells to prevent the activation of the 2C-like state transcriptional program. Methylated LINE1 recruits the methyltransferse SETDB1 and its co-factor TRIM28 through YTHDC1. Nucleolin also takes part in the silencing complex assembled over the m6A-marked LINE1. Created with BioRender.com.

**Figure 2 genes-13-00340-f002:**
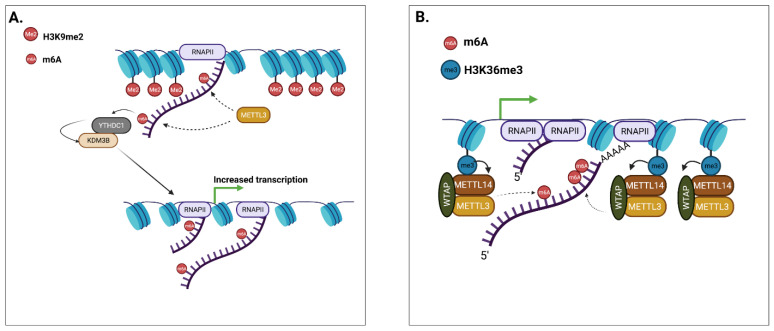
The interplay between histone modifications, gene transcription, and m6A. (**A**) METTL3-dependent methylation of mRNAs recruits YTHDC1, which in turn loads histone demethylase KDM3B to remove histone H3-lysine9 dimethylation (H3K9me2) and promote gene transcription. (**B**) Histone H3-lysine36 trimethylation (H3K36me3) recruits the m6A methyltransferase complex onto accessible chromatin, and promotes the methylation of newly transcribed mRNAs. Created with BioRender.com.

**Table 1 genes-13-00340-t001:** The interplay between m6A and chromatin accessibility.

Cells	Writer/Reader	Target RNA	Histone Signature	Effect of Writer or Reader KOon Histone Signature	Affected Genomic Loci	Biological Function of m6A	Refs.
mNSc	METTL14	CBP, P300 mRNAs	H3K27ac H3K27me3	Increased	Proliferation/differentiation gene sets	Regulating self-renewal and differentiation	[62]
mESc	METTL3, METTL14, YTHDC1	carRNAs, LINE1	H3K4me3 H3K27ac	Increased	eRNAs, paRNAs, repeat RNAs	Repressing transcription at intergenic regions	[79]
mESc	YTHDC1, METTL3	IAPs, LINE1	H3K9me3	Decreased	IAP repeats, Dux1 locus	Repressing transcription of 2C-like state genes	[80,81]
mESc	METTL3, YTHDC1	IAPs	H3K9me3 H3K20me3	Decreased	IAP repeats	Maintaining heterochromatin at repeat elements	[82]
mESc	METTL3, METTL14, YTHDF2	IAPs	H3K9me3	No significant change	ERVs	Repressing transcription of ERVs	[83]
mESC	METTL3, METTL4	MSR LINE1	H3K9me3	No significant change	MSR LINE1	Promoting the formation of RNA:DNA hybrids	[84]
HEK293, mESc	METTL3, YTHDC1	mRNAs	H3K9me2	Decreased	Gene bodies	Promoting co-transcriptional histone de-methylation and transcriptional output	[85]
mESc	METTL14	MYC ACTB	H3K36me3	No significant change	Gene rich regions	Methylate nascent RNAs	[86]
THP1	YTHDF2	KDM6B mRNA	H3K27me3	Decreased	IL-6, IL-12B, CCL22, ICAM1	Restraining the expression of pro-inflammatory cytokines	[87]

eRNAs: enhancer RNAs; paRNAs: promoter-associated RNAs; IAP: intracisternal A particle; LINE1: long interspersed nuclear elements; MSR: major satellite repeats; ERVs: endogenous retroviruses.

## Data Availability

Not applicable.

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
