# Peer review of "Driving Chromatin Organisation through N6-methyladenosine Modification of RNA: What Do We Know and What Lies Ahead?"

_genes, 2022, doi:10.3390/genes13020340_

Round 1

Reviewer 1 Report

In this review Selmi et al. present a very nice and compact review on the role of N6-methyladenosine modifications in modulating the chromatin organization. Authors have reviewed exciting findings in this field and this will be of interest for researchers investing 'epitranscriptomics' and 'epigenetics'. However there are few suggestions I would like to mention.

  1. Authors should provide a table listing the role of N6 methyl modification in driving chromatin organization which in turn guides various biological programs especially in disease. Highlighting the potential mechanisms and the target genes involved might be interesting. I do see this mentioned in the text but I would recommend to make this information visually appealing e.g. a table may be.
  2. A brief section focusing on the methods for m6A detection including chromatin organization might be interesting.
  3. In addition to lncRNAs, do other non-coding RNAs e.g. miRNAs play an important in this phenomenon.

Reviewer 2 Report

This review article discusses the recent findings of the interplay between the epigenome and the epitranscriptome, in particular the m6A RNA modification. The topic is interesting and timely, even though a recent review from Kan et al. (Kan et al. TIG 2021) already nicely described this interplay.

Overall the review is comprehensive and includes (almost) all the recent work. While I enjoyed reading it I found that in several instances the information is incorrect or imprecise. There is a lack of rigor that should be carefully addressed in a revised version. I listed below some of these points.

Lanes 54-55: “direct contact between the RNA Pol II and the MTC..”. This is not correct. Ref 12 shows that MTC can pull down Pol II but this is not a demonstration that they interact directly.

Lane 75: “HNRNPC (28)”. This is the wrong reference as it refers to hnRNPG. And only IGF2BPs have been linked to RNA stabilization.

Lane 91-93: this sentence is confusing. It’s not clear whether it is the FTO KO that prevents tumour formation or FTO itself.

Lanes 109-112: This sentence is very long and not clear.

Lane 119: 65 is a wrong reference. It should be 66.

Lanes 141-142: what does “piwi sequences” mean?

Lanes 163-168: This sentence is very long and not clear.

Lane 185: YY1 is not a histone acetyltransferase

Lane 247: HP1 is not a methyltransferase

Lane 333: “a group of three studies reported the direct m6A dependent regulation of chromatin accessibility and transcriptional output in mESc.”

The first example mentioned deals with the H3K36me3 recruiting METTL14. But in this case m6A has no influence on chromatin or transcription, it’s the other way around. Therefore this example is not valid and should be removed from this section. (in addition the panel A and B are inverted with regards to the description in the text).

The ref Wu et al, Sci adv 2020 about the interplay between m6A and H3K27me3 could also be discussed.

There are several typos (e.g. lanes 62, 151, 292, 412) and incorrect grammatical sentences (random use of past or present forms). The review would benefit from editing by a native English speaker.

Round 2

Reviewer 1 Report

Authors have satisfactorily responded to all comments.

Reviewer 2 Report

The authors revised their manuscript as requested and is now significantly improved.